# PHYSICS INFORMED DEEP KERNEL LEARNING

## ABSTRACT

Deep kernel learning is a promising combination of deep neural networks and nonparametric function estimation. However, as a data driven approach, the performance of deep kernel learning can still be restricted by scarce or insufficient data, especially in extrapolation tasks. To address these limitations, we propose Physics Informed Deep Kernel Learning (PI-DKL) that exploits physics knowledge represented by differential equations with latent sources. Specifically, we use the posterior function sample of the Gaussian process as the surrogate for the solution of the differential equation, and construct a generative component to integrate the equation in a principled Bayesian hybrid framework. For efficient and effective inference, we marginalize out the latent variables in the joint probability and derive a simple model evidence lower bound (ELBO), based on which we develop a stochastic collapsed inference algorithm. Our ELBO can be viewed as a nice, interpretable posterior regularization objective. On synthetic datasets and real-world applications, we show the advantage of our approach in both prediction accuracy and uncertainty quantification.

## 1 Introduction

Deep kernel learning (Wilson et al., 2016a) uses deep neural networks to construct kernels for nonparametric function estimation (*e.g.,* Gaussian processes (Williams and Rasmussen, 2006)) and unifies both the expressive power of neural networks and self-adaptation of nonparametric function learning. Many applications have shown that deep kernel learning substantially outperforms the conventional shallow kernel learning (*e.g.,* RBF). Compared to standard neural networks, deep kernel learning enjoys closed-form posterior distributions and hence is more convenient for uncertainty quantification and reasoning, which is important for decision making.

Nonetheless, as a data driven approach, the performance of deep kernel learning can still be restricted by scarce data, especially when the training samples are insufficient to reflect the complexity of the system (that produced the data) or the test points are far away from the training set, *i.e.,* extrapolation. On the other hand, physics knowledge, expressed as differential equations, are used to build physical models for various science and engineering applications (Lapidus and Pinder, 2011). These models are meant to characterize the underlying mechanism (*i.e.,* physical processes) that drives the system (*e.g.,* how the heat diffuses across the spatial and temporal domains) and are much less restricted by data availability: they can make accurate predictions even without training data, *e.g.,* the landing of Curiosity on Mars and flight of Voyager 1.

Therefore, we consider integrating physics knowledge into deep kernel learning to further improve its performance in prediction and uncertainty quantification, especially for scarce data and extrapolation tasks. Our work is enlightened by the recent Physics Informed Neural Networks (PINNs) (Raissi et al., 2019). However, there are two substantial differences. First, PINNs require the form of the differential equations to be fully specified. We allow the equations to include unknown latent sources (functions), which is of often the case in practice. Second, we integrate the differential equations with a principled Bayesian manner to pursue better calibrated posterior estimations.

Specifically, we use the posterior sample of the Gaussian process (GP), which is a random function, as the surrogate of the solution of the differential equation. We then apply the differential operators in the equation to obtain the sample of the latent source (function), for which we assign another GP prior. To ensure the sampling procedure is valid, we use the symmetric property of the Gaussian distribution to sample a set of virtual observations $\{0\}$, which is computationally equivalent to placing the GP prior with zero mean function over the latent source. The sampling procedure constitutes a generative component and ties to the original deep kernel model in the Bayesian hybrid framework (Lasserre

et al., 2006). For efficient and high-quality inference, we marginalize out all the latent variables in the joint distribution to avoid approximating their complex posteriors. Then we use Jensen's inequality to derive a simple model evidence lower bound (ELBO), based on which we develop a stochastic collapsed inference algorithm. The ELBO can be further explained as a soft posterior regularization objective (Ganchev et al., 2010), regularized by physics.

For evaluation, we examined our physics informed deep kernel learning (PI-DKL) in both simulation and real-world applications. On synthetic datasets based on two commonly used differential equations, PI-DKL outperforms the standard deep kernel learning, shallow kernel learning, and latent force models (LFM) that combine the physics via kernel convolution, in both ground-truth function recovery and prediction uncertainty, especially in the case of extrapolation. We then examined PI-DKL in four real-world applications. PI-DKL consistently improves upon the competing approaches in prediction error and test log-likelihood. We applied PI-DKL for a nonlinear differential equation where LFM is infeasible. PI-DKL significantly outperforms standard deep/shallow kernel learning methods.

## 2   Background

**Gaussian Process and Kernel Learning**. The Gaussian process (GP) is the most commonly used nonparametric function prior for kernel learning. Suppose we aim to learn a function $f : \mathbb{R}^d \to \mathbb{R}$ from a training set $\mathcal{D} = (\mathbf{X}, \mathbf{y})$, where $\mathbf{X} = [\mathbf{x}_1, \cdots, \mathbf{x}_N]^\top$, $\mathbf{y} = [y_1, \cdots, y_N]^\top$, each $\mathbf{x}_n$ is a $d$ dimensional input vector and $y_n$ the observed output. To avoid under-fitting and over-fitting, we do not want to assume any parametric form of $f$. Instead, we want the complexity of $f(\cdot)$ to automatically adapt to the data. To this end , we introduce a kernel function $k(\cdot, \cdot)$ that measures the similarity of the function values in terms of their inputs. The similarity only brings in a smoothness assumption about the target function. For example, the commonly used RBF kernel, $k_{\text{RBF}}(\mathbf{x}_i, \mathbf{x}_j) = \exp(-\frac{\|\mathbf{x}_i - \mathbf{x}_j\|^2}{\eta})$, implies the function is infinitely differentiable. We then use the kernel to construct a GP prior, $f \sim \mathcal{GP}(m(\cdot), k(\cdot, \cdot))$ where $m(\cdot)$ is the mean function that is usually set to constant 0. According to GP definition, the finite projection of $f(\cdot)$ on the training inputs $\mathbf{X}$, namely $\mathbf{f} = [f(\mathbf{x}_1), \cdots, f(\mathbf{x}_N)]^\top$, follow a multivariate Gaussian distribution, $p(\mathbf{f}|\mathbf{X}) = \mathcal{N}(\mathbf{f}|\mathbf{0}, \mathbf{K})$ where $\mathbf{K}$ is the kernel matrix on $\mathbf{X}$ and each $[\mathbf{K}]_{i,j} = k(\mathbf{x}_i, \mathbf{x}_j)$. Given the function values $\mathbf{f}$, the observed outputs $\mathbf{y}$ are sampled from a noisy model. For example, when $\mathbf{y}$ are continuous, we can use the isotropic Gaussian noise model, $p(\mathbf{y}|\mathbf{f}) = \mathcal{N}(\mathbf{y}|\mathbf{f}, \tau^{-1}\mathbf{I})$ where $\tau$ is the inverse variance. We can then integrate out $\mathbf{f}$ to obtain the marginal likelihood,

$$p(\mathbf{y}|\mathbf{X}) = \mathcal{N}(\mathbf{y}|\mathbf{0}, \mathbf{K} + \tau^{-1}\mathbf{I}). \tag{1}$$

To learn the model, we can maximize the likelihood to estimate the kernel parameters and the inverse variance $\tau$. According to the GP prior, given a new input $\mathbf{x}^*$, the posterior (or predictive) distribution of the output $f(\mathbf{x}^*)$ is conditional Gaussian,

$$p(f(\mathbf{x}^*)|\mathbf{x}^*, \mathbf{X}, \mathbf{y}) = \mathcal{N}(f(\mathbf{x}^*)|\mu(\mathbf{x}^*), v(\mathbf{x}^*)), \tag{2}$$

where $\mu(\mathbf{x}^*) = \mathbf{k}_*^\top (\mathbf{K} + \tau^{-1}\mathbf{I})^{-1}\mathbf{y}$, $v(\mathbf{x}^*) = k(\mathbf{x}^*, \mathbf{x}^*) - \mathbf{k}_*^\top (\mathbf{K} + \tau^{-1}\mathbf{I})^{-1}\mathbf{k}_*$ and $\mathbf{k}_* = [k(\mathbf{x}^*, \mathbf{x}_1), \cdots, k(\mathbf{x}^*, \mathbf{x}_N)]^\top$.

**Deep Kernel Learning**. While GP priors with shallow kernels (*e.g.,* RBF and Matern) have achieved a great success in many applications, these shallow structures can limit the expressiveness of the kernels in estimating highly complicated functions, *e.g.,* sharp discontinuities and high curvatures. To address this problem, Wilson et al. (2016a) proposed to construct deep kernels with neural networks. Specifically, they first choose a shallow kernel as the base kernel. Each input is first fed into a neural network (NN), and the NN outputs are then fed into the base kernel to compute the final kernel function value. Take RBF as an example of the base kernel, we can construct a deep kernel by

$$k_{\text{DEEP}}(\mathbf{x}_i, \mathbf{x}_j) = k_{\text{RBF}}(\text{NN}(\mathbf{x}_i), \text{NN}(\mathbf{x}_j)). \tag{3}$$

Note that the NN weights now become the kernel parameters. We can then use the deep kernel to construct a GP prior for nonparametric function estimation. The likelihood and predictive distribution have the same forms as in (1) and (2).

## 3   Model

By using deep neural networks to construct highly expressive kernels, deep kernel learning greatly enhances the capability of estimating complicated functions, and meanwhile inherits the self-adaptation of the nonparametric function learning and convenient posterior inference. However, as a purely

data-drive approach, deep kernel learning can still suffer from data scarcity, especially when the training examples are inadequate to reflect the complexity of the underlying mapping and when the test points are distant from all the training samples, *i.e.,* extrapolation. To overcome this limitation, we propose PI-DKL, a physics informed deep kernel learning model that exploit physics prior knowledge to improve the function learning and uncertainty reasoning. Our model is presented as follows.

## 3.1 Physics Informed Deep Kernel Learning

We assume that in general, the physics is described by a differential equation of the following form,

$$\psi[f(\mathbf{x})] = g(\mathbf{x}) \tag{4}$$

where $\psi$ is a functional that combines a set of differential operators, $f(\mathbf{x})$ is the target (or solution) function we want to estimate from the training dataset $\mathcal{D} = (\mathbf{X}, \mathbf{y})$, and $g(\mathbf{x})$ is a latent source whose form is unknown. Note that the functional $\psi[\cdot]$ may include unknown parameters as well. One example is $\psi[f(x)] = \frac{\mathrm{d}f(x)}{\mathrm{d}x} + \alpha f(x) - \beta$, where the input $\mathbf{x}$ is a scalar, and $\alpha$ and $\beta$ are unknown parameters. This functional represents a linear operator. Another commonly seen example is from the viscous version of Burger's equation (Olsen-Kettle, 2011), $\psi[f(\mathbf{x})] = \frac{\partial f(\mathbf{x})}{\partial x_1} + f(\mathbf{x})\frac{\partial f(\mathbf{x})}{\partial x_2} - v\frac{\partial^2 f(\mathbf{x})}{\partial x_2^2}$, where $\mathbf{x} = [x_1, x_2]^\top$, $x_1$ is the spatial variable, $x_2$ the time variable, and $v$ the unknown viscosity parameter. This functional includes a nonlinear operator, $f(\mathbf{x})\frac{\partial f(\mathbf{x})}{\partial x_2}$.

To incorporate the physics knowledge in (4), we propose a hybrid of conditional and generative models based on the general framework proposed by Lasserre et al. (2006). The conditional component is the standard deep-kernel GP that given the training inputs $\mathbf{X}$, samples the (noisy) output observations $\mathbf{y}$, and the probability $p(\mathbf{y}|\mathbf{X})$ is given in (1). The generative component fulfills another GP prior over the latent source $g(\cdot)$, but avoids the double prior problem to ensure a valid joint Bayesian model for posterior inference. Coupled with the differential operators, the generative component regularizes and guides the deep kernel learning of the $f(\cdot)$. The graphical illustration of PI-DKL is shown in Fig. 1 of the supplementary document.

Specifically, to consider a GP prior over $g(\cdot)$, we first sample a finite set of input locations $\mathbf{Z} = [\mathbf{z}_1, \ldots, \mathbf{z}_m]^\top$ (we will discuss the choice of $p(\mathbf{Z})$ later). Then the projection of $g(\cdot)$ on $\mathbf{Z}$ follows a multivariate Gaussian distribution,

$$p(\mathbf{g}|\mathbf{Z}) = \mathcal{N}(\mathbf{g}|\mathbf{0}, \boldsymbol{\Sigma}) \tag{5}$$

where $\mathbf{g} = [g(\mathbf{z}_1), \ldots, g(\mathbf{z}_m)]^\top$, $[\boldsymbol{\Sigma}]_{ij} = \kappa(\mathbf{z}_i, \mathbf{z}_j)$ and $\kappa(\cdot, \cdot)$ is another kernel.

Next, we link the GP model of the target $f(\cdot)$ to the latent source $g(\cdot)$ via the differential equation (4). Our key idea is that from the GP posterior (2), we can construct a sample of the target function[1], $f(\cdot) = \mu(\cdot) + \epsilon\sqrt{v(\cdot)}$, where $\epsilon \sim \mathcal{N}(\epsilon|0, 1)$, $\mu(\cdot)$ and $\sqrt{v(\cdot)}$ are the posterior mean and standard deviation functions. While this is a random function (due to $\epsilon$), it has a closed form and we can apply the functional $\psi$ to obtain the sample of $g(\cdot)$,

$$g(\cdot) = h(\cdot, \epsilon) = \psi[\mu(\cdot) + \epsilon\sqrt{v(\cdot)}]. \tag{6}$$

Therefore, to sample $\mathbf{g}$ — the values of $g(\cdot)$ on $\mathbf{Z}$, we can first sample a standard Gaussian white noise $\epsilon$, and then sample from

$$p(\mathbf{g}|\epsilon, \mathbf{X}, \mathbf{y}) = \prod_{j=i}^{m} \delta\left(\widetilde{g}_j - h(\mathbf{z}_j, \epsilon)\right), \tag{7}$$

where $\widetilde{g}_j = g(\mathbf{z}_j)$, and $\delta(\cdot)$ is the Dirac delta prior. Note that we can also directly view $\mathbf{g}$ as a transformation of the Gaussian noise $\epsilon$ and derive the marginal distribution $p(\mathbf{g}|\mathbf{X}, \mathbf{y})$ (see the discussion in the supplementary material), which, however, is much more difficult to compute.

Now, we want to tie the GP prior for $g(\cdot)$ in (5) to the samples $\mathbf{g}$ generated from the GP model of the target function $f(\cdot)$, *i.e.,* through (7). In this way, the learning of $f(\cdot)$ can be guided or regularized by the differential equation (4). However, directly multiplying (5) and (7) is problematic, because $\mathbf{g}$ will have double priors and the sampling procedure is invalid — if $\mathbf{g}$ is sampled from (5), it cannot

---

[1]In computational physics, this is viewed as a surrogate for the solution function of the differential equation.

be sampled again from (7), and vice versa. To ensure our model is a valid probabilistic model for posterior inference, we utilize the symmetric property of the Gaussian distribution,

$$p(\mathbf{g}|\mathbf{Z}) = \mathcal{N}(\mathbf{g}|\mathbf{0}, \boldsymbol{\Sigma}) = \mathcal{N}(\mathbf{0}|\mathbf{g}, \boldsymbol{\Sigma}) = p(\mathbf{0}|\mathbf{g}, \mathbf{Z}). \tag{8}$$

We can see that placing a (finite) GP prior over $g(\cdot)$ is equivalent to sampling a set of virtual observations $\mathbf{0}$, due to the symmetry of the Gaussian distribution. Therefore, we can turn the GP prior of the latent source to a generative component that samples a set of virtual observations $\mathbf{0}$. From the computational perspective, they are totally equivalent. However, the sampling procedure now becomes valid — we first sample $\mathbf{g}$ from (7), and then sample $\mathbf{0}$ from (8). Note that the virtual observations $\mathbf{0}$ come from the zero-mean function of the GP prior of $g(\cdot)$. We can use different virtual observations by choosing a nonzero mean function.

Finally, we combine the conditional model and the generative model (see (1), (7) and (8)) to obtain a joint probability distribution,

$$p(\mathbf{y}, \mathbf{0}, \mathbf{Z}, \epsilon, \mathbf{g}|\mathbf{X}) = p(\mathbf{y}|\mathbf{X})p(\mathbf{Z})p(\epsilon)p(\mathbf{g}|\epsilon, \mathbf{X}, \mathbf{y})p(\mathbf{0}|\mathbf{g}, \mathbf{Z})$$
$$= \mathcal{N}(\mathbf{y}|\mathbf{0}, \mathbf{K} + \tau^{-1}\mathbf{I})p(\mathbf{Z})\mathcal{N}(\epsilon|0, 1) \prod_{j=1}^{m} \delta\left(\widetilde{g}_j - h(\mathbf{z}_j, \epsilon)\right) \mathcal{N}(\mathbf{0}|\mathbf{g}, \boldsymbol{\Sigma}). \tag{9}$$

The choice of $p(\mathbf{Z})$ is flexible. If we have no knowledge about the input distribution, we can use a uniform distribution for the bounded domain, and for unbounded domains we can use a wide Gaussian distribution with zero mean or uniform distribution on a region large enough to cover our interested predictions.

# 4 Algorithm

## 4.1 Stochastic Collapsed Inference

We now present the model inference algorithm. The exact posterior of the latent random variables $\mathbf{Z}$, $\epsilon$, and $\mathbf{g}$ in (9) are infeasible to calculate because they are coupled in kernels and differential operators. While we can use variational approximations, they will introduce extra variational parameters, complicate the optimization and affect the integration of the physics knowledge. Therefore, we marginalize out all the latent variables to conduct collapsed inference to avoid approximating their complex posteriors. Specifically, we derive that $p(\mathbf{y}, \mathbf{0}|\mathbf{X}) = p(\mathbf{y}|\mathbf{X})p(\mathbf{0}|\mathbf{y}, \mathbf{X})$, where

$$p(\mathbf{0}|\mathbf{y}, \mathbf{X}) = \int p(\mathbf{Z})p(\epsilon)p(\mathbf{g}|\epsilon, \mathbf{X}, \mathbf{y})p(\mathbf{0}|\mathbf{g}, \mathbf{Z})\mathrm{d}\mathbf{Z}\mathrm{d}\epsilon\mathrm{d}\mathbf{g}$$

$$= \mathbb{E}_{p(\mathbf{Z})}\mathbb{E}_{p(\epsilon)}\Big[\int \delta(\mathbf{g} - \mathbf{h})\mathcal{N}(\mathbf{0}|\mathbf{g}, \boldsymbol{\Sigma})\mathrm{d}\mathbf{g}\Big] = \mathbb{E}_{p(\mathbf{Z})}\mathbb{E}_{\mathcal{N}(\epsilon|0,1)}\left[\mathcal{N}\left(\mathbf{h}(\mathbf{Z}, \epsilon)|\mathbf{0}, \boldsymbol{\Sigma}\right)\right]. \tag{10}$$

where $\mathbf{h}(\mathbf{Z}, \epsilon) = [h(\mathbf{z}_1, \epsilon), \dots, h(\mathbf{z}_m, \epsilon)]^\top$. Note that $h(\cdot, \cdot)$ is defined in (6).

To allow us to adjust the importance of the generative component and so the influence of the physics during training, we weight the likelihood of the generative component by a free hyper-parameter $\gamma \geq 0$. The weighted marginal likelihood (Warm, 1989; Hu and Zidek, 2002) is

$$p_\gamma(\mathbf{y}, \mathbf{0}|\mathbf{X}) = p(\mathbf{y}|\mathbf{X})p(\mathbf{0}|\mathbf{X}, \mathbf{y})^\gamma. \tag{11}$$

Our inference is to maximize the log weighted marginal likelihood to optimize the kernel parameters in $k_{\mathrm{DEEP}}(\cdot, \cdot)$ and $\kappa(\cdot, \cdot)$, the inverse noise variance $\tau$ and unknown parameters in the differential equation, $\log p_\gamma(\mathbf{y}, \mathbf{0}|\mathbf{X}) = \log\left(\mathcal{N}(\mathbf{y}|\mathbf{0}, \mathbf{K} + \tau^{-1}\mathbf{I})\right) + \gamma \log\left(\mathbb{E}_{p(\mathbf{Z})}\mathbb{E}_{\mathcal{N}(\epsilon|0,1)}\left[\mathcal{N}\left(\mathbf{h}(\mathbf{Z}, \epsilon)|\mathbf{0}, \boldsymbol{\Sigma}\right)\right]\right)$. However, the log likelihood is infeasible to compute due to the intractable expectation inside the logarithm. To address this problem, we use Jensen's inequality on the log function to obtain a model evidence lower bound (ELBO), $\mathcal{L} \leq \log p_\gamma(\mathbf{y}, \mathbf{0}|\mathbf{X})$, where

$$\mathcal{L} = \log\left(\mathcal{N}(\mathbf{y}|\mathbf{0}, \mathbf{K} + \tau^{-1}\mathbf{I})\right) + \gamma \cdot \mathbb{E}_{p(\mathbf{Z})}\mathbb{E}_{\mathcal{N}(\epsilon|0,1)}\left[\log\left(\mathcal{N}\left(\mathbf{h}(\mathbf{Z}, \epsilon)|\mathbf{0}, \boldsymbol{\Sigma}\right)\right)\right]. \tag{12}$$

While $\mathcal{L}$ is still intractable, it is straightforward to maximize $\mathcal{L}$ with stochastic optimization. Each time, we generate a sample of the input locations from $p(\mathbf{Z})$ and the noise from $\mathcal{N}(\epsilon|0, 1)$, denoted by $\widetilde{\mathbf{Z}}$ and $\widetilde{\epsilon}$. We then obtain $\widetilde{\mathcal{L}} = \log\left(\mathcal{N}(\mathbf{y}|\mathbf{0}, \mathbf{K} + \tau^{-1}\mathbf{I})\right) + \gamma \log\left(\mathcal{N}(\mathbf{h}(\widetilde{\mathbf{Z}}, \widetilde{\epsilon})|\mathbf{0}, \boldsymbol{\Sigma})\right)$, a unbiased stochastic estimation of $\mathcal{L}$. We calculate $\nabla\widetilde{\mathcal{L}}$ as an unbiased stochastic gradient of $\mathcal{L}$, with which we can use any stochastic optimization to estimate the model parameters. While $\mathbf{h}(\cdot, \cdot)$ couples the

deep kernels and complex operators in $\psi$, it is differentiable and we can use automatic differentiation libraries to calculate the stochastic gradient conveniently.

The ELBO $\mathcal{L}$ in (12) is the GP log marginal likelihood plus an extra term, $\mathbb{E}_{p(\mathbf{Z})}\mathbb{E}_{\mathcal{N}(\epsilon|0,1)}\left[\log\left(\mathcal{N}\left(\mathbf{h}(\mathbf{Z},\epsilon)|\mathbf{0},\boldsymbol{\Sigma}\right)\right)\right]$. Each element of $\mathbf{h}$ is obtained by applying the functional $\psi$ on the posterior sample of $f(\cdot)$ (see (6)). Jointly maximizing this term in $\mathcal{L}$ encourages that all the possible latent source values (at $m$ locations) obtained from the GP posterior function $f(\cdot)$ (through the equation) should be considered as the samples of another GP. This can be viewed as a soft constraint over the posterior function of the GP. Therefore, our ELBO is also a posterior regularization objective (Ganchev et al., 2010), and our inference estimates the standard deep-kernel GP model with a soft regularization on its posterior distribution.

### 4.2 Algorithm Complexity

The time complexity for the inference of our model is $\mathcal{O}(N^3 + m^3)$, because it involves the calculation for two GPs: one is the standard model, and the other is in the generative component. The time complexity for prediction is still $\mathcal{O}(N^3)$. The space complexity is $\mathcal{O}(N^2 + m^2)$, including the storage of the kernel matrices of the two GPs.

## 5 Related Work

An influential work, physics informed neural networks (PINNs) (Raissi et al., 2019), were recently proposed to train neural networks that respect physical laws. The key idea is to use neural networks as a surrogate for the solution of the (partial) differential equation, and minimize the NN loos plus the residual error on a set of randomly collected collocation points in the input domain. Research along this line are quickly growing: (Mao et al., 2020; Jagtap et al., 2020; Zhang et al., 2020; Chen et al., 2020; Pang et al., 2019), to name a few. While our work is enlightened by PINNs, there are several substantial differences. First, PINNs demand the form of the PDE is fully specified, *i.e.,* $\psi[f(\mathbf{x})] = 0$, while we assume there can be some unknown source (function), $\psi[f(\mathbf{x})] = g(\mathbf{x})$. Thus, our work is to exploit incomplete physics knowledge. Second, we use the posterior of the deep-kernel GP to construct a random surrogate for the PDE solution, and cast the integration of the physics into a principled Bayesian framework to enable posterior inference and uncertainty quantification, while PINNs only conduct point estimations. Our experiments show that the incomplete physics knowledge can also improve uncertainty quantification. Note that in the mean time, Zhang et al. (2019) combined polynomial chaos (Xiu and Karniadakis, 2002) and dropout (Gal and Ghahramani, 2016) to estimate the total uncertainty for PINNs with stochastic PDEs

Many works have used GPs to model or learn physical systems (Graepel, 2003; Lawrence et al., 2007; Gao et al., 2008; Alvarez et al., 2009; 2013; Raissi et al., 2017). For example, Graepel (2003) uses GPs to solve the linear equation given observed noisy sources. He first defines the kernel for the solution function with which to derive the kernel for the source function. The kernel parameters are then estimated from the noisy source data, given which the solution can be predicted. Raissi et al. (2017) assume both the noisy forces and solutions are observed, and they jointly model these examples in one single GP with a heterogeneous block covariance matrix. Latent force models (LFM) (Alvarez et al., 2009) make the same assumption about the differential equations as our work. LFMs convolve the Green's function of the equation with the kernel for the latent source to obtain the kernel for the target, and then learn the kernel parameters from data. While LFMs enable a hard encoding of physics, they rely on an analytical Green's function, which is not available for many equations. In addition, LFMs construct shallow kernels, which can be less expressive as deep kernels. Other works include (Calderhead et al., 2009; Barber and Wang, 2014; Macdonald et al., 2015; Heinonen et al., 2018; Wenk et al., 2020) *etc.* They mainly focus on estimating parameters/operators in ODEs without latent functions/sources.

## 6 Experiments

### 6.1 Simulation

We first examined if PI-DKL can improve extrapolation with right physics knowledge. We generated two synthetic datasets. The first dataset, *1stODE*, was simulated from a first-order Ordinary Differential Equation (ODE), $\frac{\partial f(t)}{\partial t} + B \cdot f(t) - D = g(t)$ where $B = D = 1$, $g(t) = \sin(2\pi t)\exp(-t \cdot$ and the initial condition $f(0) = 0.1$. We set the time domain $t \in [0, 1]$. We ran the finite difference algorithm (Mitchell and Griffiths, 1980) to obtain the accurate solution. We chose $1,001$ equally spaced time points ($t_0 = 0, t_{1000} = 1$) and their solution values as the dataset. The second dataset, *1dDiffusion*, was simulated from a diffusion equation with one dimensional spatial domain,

$\frac{\partial f(x,t)}{\partial t} - \alpha \frac{\partial f^2(x,t)}{\partial x^2} = g(x,t)$ where $\alpha = 10$, $g(x,t) = 0$ and the initial condition $f(x,0)$ is a square wave. We set the domain $(x,t) \in [0,1] \times [0,1]$. We ran a numerical solver to obtain the accurate solution. Then we discretized the entire spatial and time domain into a $48 \times 101$ grid with equal spacing in each dimension. We retrieved the grid points and their solution values as our dataset.

**Competing methods**. We compared with (1) shallow kernel learning (SKL) with SE-ARD kenrel, (2) deep kernel learning (DKL), and (3) LFM, which uses SE-ARD for the latent source, and then convolves it with Green's function to obtain the kernel for the target function. To construct a deep kernel, we followed (Wilson et al., 2016a) to feed the input variables to a (deep) neural network (NN) and calculated the RBF kernel over the neural network outputs (see (3)). Across our experiments, we used a 5-layer NN, with 20 nodes in each hidden and output layer. We used $\tanh(\cdot)$ as the activation function. For our method PI-DKL, we used the same deep kernel for the target function. As in LFM, we used SE-ARD kernel for the latent source. We set the number of virtual observations $m = 10$ for the generative component, and uniformly sampled the input locations from the entire domain (see (12)). We chose the weight of the generative component $\gamma$ from $\{0.01, 0.05, 0.1, 0.5, 1, 2, 5, 10\}$. For both LFM and PI-DKL, the parameters of differential equations are unknown. All the methods were implemented with TensorFlow (Abadi et al., 2016). For our method, we used ADAM (Kingma and Ba, 2014) for stochastic inference. We ran 10K epochs to ensure convergence. For the other methods, we used L-BFGS for optimization and set the maximum number of iterations to 5K.

For *1stODE* , we used the first 101 samples ($t_i \in [0,0.1]$) for training, and the remaining 900 samples ($t_i \in (0.1, 1]$) for test. We show the posterior distribution of the functions learned by all the methods and the ground-truth in Fig. 1. We can see that the predictions of SKL and DKL are largely biased when the test points are far from the training region $[0, 0.1]$. On average, DKL obtains better accuracy than SKL. The root-mean-square errors (RMSEs) are {DKL:0.21, SKL:0.25}. As a comparison, the posterior means of LFM and PI-DKL are much closer to the ground-truth in the test region, and the RMSEs are {LFM: 0.09, PI-DKL: 0.04}, showing the benefit of the physics. However, LFM is quite unstable in extrapolation: the farther away the test area, the more fluctuating the prediction. By contrast, PI-DKL obtains much smoother curves that are even closer to the ground-truth, and smaller posterior variances in the test region. Hence, it shows that the LFM kernel obtained from shallow kernel convolution is less expressive/powerful than the regularized deep kernel in PI-DKL. Note that unlike SKL/DKL, both LFM and PI-DKL estimated nontrivial posterior variances (*i.e.,* not extremely close to 0) in the training region, implying that the physics also helps prevent overfitting.

Since for diffusion equations, LFM cannot derive the kernel for time variable $t$, for a fair comparison on *1dDiffusion*, we fixed $t = 0.5$ and used the 48 spatial points as the training inputs. We then evaluated the posterior distribution of the function values at all the grid points ($48 \times 101$) in the entire domain. We report the absolute difference between the posterior mean and ground-truth in Fig. 2a-d. We can see that the prediction errors of SKL/DKL are close to 0 (dark colors) in regions close to the training data ($t = 0.5$). However, when the test points are getting far away, say, close to the boundary ($t = 0$ or 1), the error grows significantly (see the bright colors). Overall, DKL still achieves smaller extrapolation error than SKL, implying an advantage of using more flexible deep kernels. From Fig. 2c, we can see that while LFM misses the time information, it still exhibits better extrapolation results, as compared with SKL/DKL, showing the benefit of the physics . PI-DKL achieves even smaller prediction error (*i.e.,* darker) when $t$ is away from the training time point and exhibits even best extrapolation performance. The RMSEs of all the methods are {SKL: 0.18, DKL: 0.11, LFM: 0.09, PI-DKL:0.07}. We also report the predictive standard deviation (PSD) of each method in Fig. 2e-f. We can see that the PSDs of SKL/DKL are both close to 0 in the training region, and quickly increase when the inputs move away (on average DKL shows smaller PSDs and smoother changes). By contrast, LFM and PI-DKL obtain PSDs quite uniformly across the entire domain and less than SKL/DKL. It means that the physics knowledge help inhibit overfitting and reduce the uncertainty in extrapolation. Compared with LFM, PI-DKL obtains even smaller PSDs (darker color) across the domain, showing even smaller uncertainty in extrapolation.

## 6.2 Real-World Applications

**Metal Pollution in Swiss Jura.** Next, we evaluated PI-DKL in real-world applications. We examined the predictive performance in terms of normalized RMSE (nRMSE) and test log-likelihood (LL). Due to the space limit, the test LL results are provided in the supplementary material. We first considered predicting the metal concentration in Swiss Jura. The data were collected from 300 locations in a 14.5 km$^2$ region (`https://rdrr.io/cran/gstat/man/jura.html`). The diffusion of the metal concentration is naturally modelled by a diffusion equation with the two-dimensional

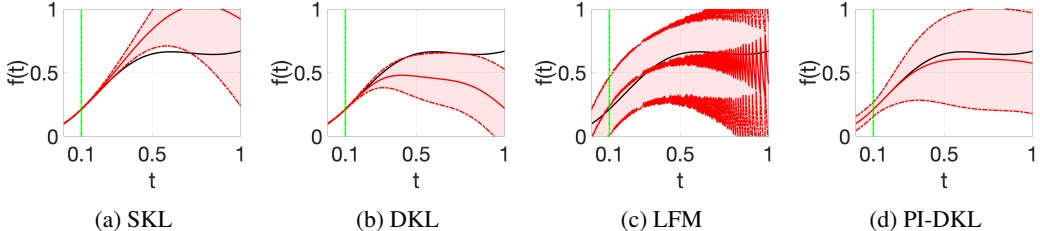

Figure 1: The posterior distribution of the learned solution functions on *1stODE*. The red lines in the middle are the posterior means and the red dashed lines on the boundary of the shaded region the posterior mean plus/minus one posterior standard deviation. The black line is the ground-truth solution. The training inputs stay in $[0, 0.1]$ (left to the green line).

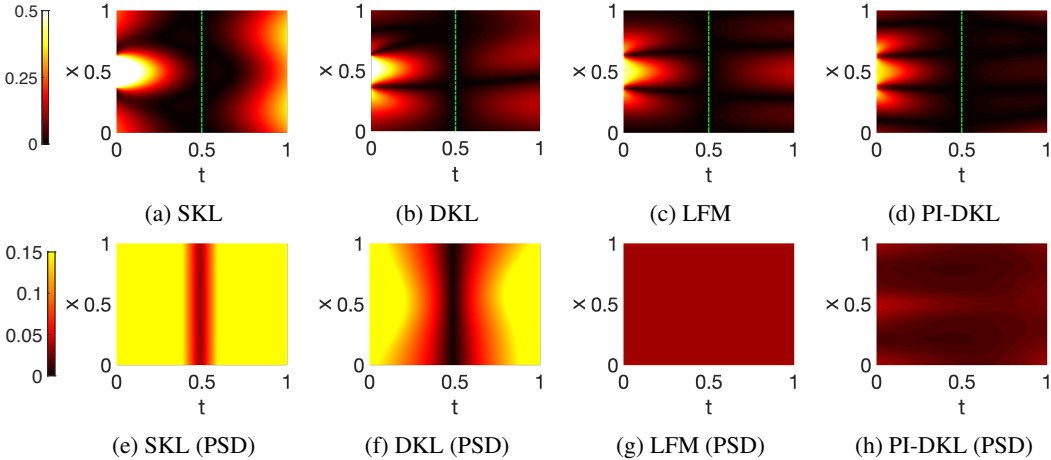

Figure 2: The absolute value of the difference between the posterior mean and the ground-truth (1st row) and posterior standard deviation (2nd row) on *1dDiffusion*. The training examples stay on $t = 0.5$ (the green line).

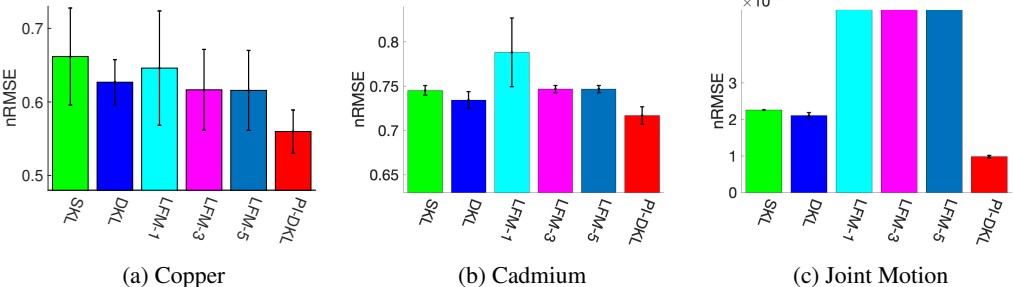

Figure 3: Metal concentration prediction in Swiss Jura (a, b) and joint angle prediction in motion capture (c). The results are averaged over 5 runs. The normalized root-mean-square error (nRMSE) in each run is computed by normalizing the RMSE by the mean of the test outputs.

spatial domain, $\frac{\partial f(x_1, x_2, t)}{\partial t} = \alpha(\frac{\partial f^2(x_1, x_2, t)}{\partial x_1^2} + \frac{\partial f(x_1, x_2, t)}{\partial x_2^2})$, where $f(\cdot, \cdot, \cdot)$ is the concentration of the metal at a particular location and time point. However, the dataset do not include the time $t_s$ when these concentrations were measured. LFM considers the initial condition $f(x_1, x_2, 0)$ as the latent function and obtains a kernel of the locations where $t_s$ can be viewed as a kernel parameter learned from data. In our approach, we estimated the solution function at $t_s$, $h(x_1, x_2) = f(x_1, x_2, t_s)$. Hence, the equation can be viewed as $\frac{\partial h^2(x_1, x_2)}{\partial x_1^2} + \frac{\partial h^2(x_1, x_2)}{\partial x_2^2} = g(x_1, x_2)$, where the latent function $g(x_1, x_2) = \frac{1}{\alpha} \frac{\partial f(x_1, x_2, t)}{\partial t}|_{t=t_s}$. We were interested in predicting the concentration of cadium and copper. The input variables include the coordinates of the location $(x_1, x_2)$, the concentrations of {nickel, zinc} for cadmium, and {lead, nickel, zinc} for copper. For PI-DKL, we selected $m$ from $\{10, 50, 100, 200, 500\}$ for the generative component and $\gamma$ from $\{0.01, 0.05, 0.1, 0.5, 1, 2, 5, 10\}$. We normalized the training inputs and then sampled latent inputs $\mathbf{Z}$ from $\mathcal{N}(\mathbf{0}, \mathbf{I})$ in model estimation. For LFM, we varied the number of latent forces from $\{1, 3, 5\}$. We randomly selected 50 samples for training, and used the remaining 250 samples for test. We repeated the experiments for 5 times, and report the average nRMSE and its standard deviation of each method in Fig. 3a and b. PI-DKL

outperforms all the competing approaches for both prediction tasks. PI-DKL always significantly improves upon SKL and DKL ($p < 0.05$). In addition, PI-DKL significantly outperforms LFM in predicting Cadium concentration (Fig. 3b). Note that LFM does improve upon SKL in predicting Copper concentration (Fig. 3a), but not as significant as PI-DKL.

**Motion Capture.** We then looked into predicting trajectories of joints in the motion capture application. To this end, we used CMU motion capture database ( `http://mocap.cs.cmu.edu/`), from which we used the samples collected from subject 35 in the walk and jog motion lasting for 2,644 seconds. We trained all the models to predict the angles of Joint 60 along with time. We used the first order ODE in simulation to represent the physical model, based on which we ran LFM and PI-DKL. Note this physical system might be oversimplified (Alvarez et al., 2009). For LFM, we varied the number of latent forces from {1,3, 5}. Again, we randomly selected 500 samples for training and 2,000 samples for test. We repeated the experiments for 5 times and report the average nRMSE and its standard deviation in Fig. 3c. As we can see, PI-DKL improves upon all the competing methods by a large margin. Note that LFM is even far worse than SKL. This might because LFM over-exploits the over-simplified physics, which harms the prediction. By contrast, PI-DKL allows us to tune the number of virtual observations $m$ and the likelihood weight ($\gamma$ in (12)), and hence can consistently improve upon DKL.

**PM2.5 in Salt Lake City.** Second, we considered predicting the Particulate Matter (PM2.5) levels across Salt Lake City. The dataset were collected from sensors' reads at different time and locations (`https://aqandu. org/`). We chose the time range from 07/04/2018 to 07/06/2018. Following (Wang et al., 2018), we used the diffusion equation plus a source term (*i.e.,* the latent function) to

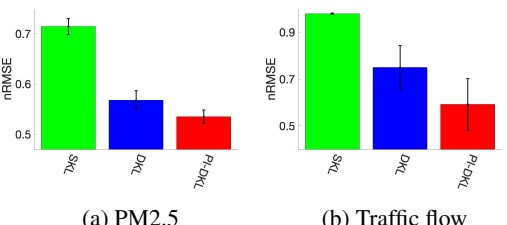

(a) PM2.5      (b) Traffic flow

Figure 4: PM2.5 and traffic flow prediction.

represent the physical model, $\frac{\partial f(x_1,x_2,t)}{\partial t} - \alpha \sum_{j=1}^{2} \frac{\partial f^2(x_1,x_2,t)}{\partial x_j^2} = g(x_1, x_2, t)$, where $f$ is the concentration level and $g$ the source term. The input variables include both the location coordinates and detailed time points. Since LFM cannot construct a full kernel of the input variables from the physics, we did not test it to avoid unfair comparisons. We trained SKL and DKL with both the spatial and time inputs. We randomly selected 500 samples for training and 2,000 samples for test. We repeated the experiments for 5 times and report the average nRMSE and its standard deviation in Fig. 4a. As we can see, with a more expressive kernel, DKL improves upon SKL significantly, and with the incorporation of the physics, PI-DKL in turn outperforms DKL significantly ($p < 0.05$).

**High-Way Traffic Flow Prediction.** Finally, we applied PI-DKL to predict the traffic flow in the interstate highway 215 across Utah state. The Utah Department of Transportation (UDOT) has installed sensors every a few miles along the high way. Each sensor counts the number of vehicles passed every minute, and sends the data back to a central database. The real time data and road conditions are available at `https://udot.iteris-pems.com/`. We used the data collected by 20 sensors continuously installed in a segment of 30 miles, and the time was chosen from 08/05/2019 to 08/11/2019. The input variables include the location coordinates of each sensor and the time of each read. Following (Nagatani, 2000), we used the Burger's equation plus a source term to describe the system, $\frac{\partial f}{\partial t} + f \cdot \sum_{j=1}^{2} \frac{\partial f}{\partial x_i} - \nu \sum_{j=1}^{2} \frac{\partial f^2}{\partial x_j^2} = g(x_1, x_2, t)$, where $f$ is the traffic flow, $\nu$ the unknown viscous coefficient, and $g$ the source term, *i.e.,* the latent function. Note that the equation is nonlinear and we do not have an analytical form of Green's function. Hence we cannot use LFM to incorporate the physics to enhance GP training. Hence we compared with SKL and DKL only. We randomly selected 500 and 2,000 samples for training and test, respectively, and repeated for 5 times. The average nRMSEs and the standard deviations are reported in Fig. 4b. As we can see, DKL significantly outperforms SKL, which demonstrates the advantage of the more expressive, deep kernel. More important, PRGP further improves upon DKL, showing that the physics incorporated by our approach indeed promotes the predictive performance.

## 7 Conclusion

We have presented PI-DKL, a physics informed deep kernel learning model that can flexibly incorporate physics knowledge from incomplete differential equations to improve function learning and uncertainty quantification. In the future work, we will extend our model with sparse approximations (Hensman et al., 2013; Wilson et al., 2016b) to exploit physics in large-scale applications.

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
