# OpenReview forum: "Physics Informed Deep Kernel Learning"
_ICLR.cc/2021/Conference — Reject_

### Official Review · AnonReviewer2 · 2020-10-21
**Sound work**

**Rating:** 7
**Confidence:** 3

**Review:**

### Summary of my understanding

The authors address the problem of function estimation given noisy observations of the function's values. The estimation is done in a non-parametric manner as posterior inference of Gaussian processes. The authors' proposal is to incorporate physics knowledge into this process. The knowledge here refers to a differential equation on the function to be estimated ($f$), and it may include unknown parameters and an unknown external force term $g$. They derive a joint distribution of 1) observation of $f$, i.e., $y$, 2) "virtual" observations $0$ of $g$, 3) latent variable $Z$ for latent force $g$, 4) randomness between $f$ and $g$, and 5) values of $g$. Afterward, they propose to perform a collapsed inference based on a lower bound of the log marginal likelihood (of observations $y$ and virtual observations $0$).

### Evaluation

The problem setting is clear. The proposed method is technically sound, and the experiments are enough convincing to show its superiority to baselines. Regardless of the common points with the latent force models [Alvarez+ 09] in terms of the problem setting, this paper proposes a somewhat new technique to perform inference under the presence of differential equation-based constraints on a target function. I think this paper is a sound work.

### Note

I think a concern lies in the paper's clear violation of the formatting instruction of ICLR 2021. Especially, the too narrow margins before and after headings seem to compress the main text, which might seem quite unfair. I delegate judgment on this regard to the chairs and for now, I decided the rating ignoring this matter.

---

> ### Author Response · Authors · 2020-11-14
> **Thanks for your comments**
>
> C: comments; R: response
>
> C1: two narrow margins before and headings.
>
> R1: Thanks for pointing out this issue. We will definitely compress the text and make the margins better fit for the original template.

---

### Official Review · AnonReviewer3 · 2020-10-23

**Rating:** 5
**Confidence:** 4

**Review:**

Post-discussion update: The authors have clarified their work considerably, and I believe the work is probably correct. However, the paper still suffers from poor presentation and poorly-motivated or justified modelling choices. The current version of the paper has not been updated, and all below issues thus stand. The paper overall presents a good idea, but I believe the authors made poor modelling choices, which led the kludgy math.

----

The paper proposes to combine deep kernel learning (DKL) with PDE/ODE prior knowledge for learning spatiotemporal systems. The idea is sound and sensible, although it represents an incremental combination of two existing techniques. The differential assumption clearly improves in spatiotemporal experiments, which also motivates the idea.

The paper has unfortunately three major mathematical errors.

First, the very first equation of the method (eq 4) is already incorrect: g(x) is not a function of input “x” but instead it’s a function of solution “f”. The paper itself acknowledges this by giving series of examples which all are functions of “f”. The following GP prior for g(x) is then placed on incorrect inputs and seems misguided or at least insufficient to model the differential. Intuitively this is also easy to see: the differential of the solution is obviously not just a function of space and time (but of solution as well).

Second, the eq 6 states that a non-linear differential of a GP is some other Gaussian process. This is incorrect: non-linear transforms of a Gaussian process does not retain Gaussianity. Since this is a central definition of the paper, the whole method is likely incorrect.

Third, the paper introduces a bizarre concept of constant zero vectors as random variables, ie. p(0|g). This is clearly wrong, and the probabilistic model is then wrong as well. While this could be easily fixed, the model is an example of GP-matching (which has known pathologies), and this has been extensively studied by many authors (eg. Wenk’19, Wenk’18, Gorbach’17, MacDonalds’15).

Given these obvious errors, the paper needs a major revision.


Technical comments:
o Eq 4 should be g(f,x), since later the \varphi is a function of f and its x-derivatives. Currently none of the examples (eg. burger) follow eq 4. Also eq6 is also wrong with same argument
o Please define (mathematically) the tackled problem and the problem domain. The paper starts by discussing classification, but at sec 3 the context suddenly changes into PDEs without the reader being informed about this.
o The paragraph “to incorporate..” is difficult to follow since its technical but does not open up the math yet. It would help to write this in more conceptual way
o It would greatly help the reader understand the method to include the sup-fig1 in the main paper
o It seems that both f and g are assumed to be separate GPs. This is unclear from text, please clarify
o eq7 is strange, since it defines a dirac between a random variable g and a sample of h. This is nonsensical.
o The 0-sampling is obviously wrong and unnecessary. Double priors are perfectly fine in Bayesian modelling, and eq9 could be defined without the 0-stuff. The double priors are called product of expert priors, see eg. MacDonalds'15 or Wenk’19. They do have their own pathologies, but these papers discuss them extensively.

---

> ### Author Response · Authors · 2020-11-14
> **Complete Misunderstanding and Wrong Comments**
>
> We are astonished that the reviewer totally misunderstood our paper, and left baseless and wrong comments.  C: comments; R: response.
>
> C1: the very first equation of the method (eq 4) is already incorrect: g(x) is not a function of input “x” but instead it’s a function of solution "f";
>
> R1: First, g(x) is a source term (see the 2nd line under eq 4). In the literature of differential equations, the source term usually represents external influence on the physical system (described by \psi[f]). Therefore, in general, g should be considered as a function of x, rather than assume a nested structure including f. Note that this is NOT a definition of ODE. Our work is NOT restricted to ODE either.
>
> Second, even we put an extra assumption that g is also a function of f, i.e., g(f, x),  because f is a function of x, g is in essence still a function of x. There is NOT any conceptual and mathematical error.
>
> Third, the prior works, including Graepel03, Raissi17, and latent force models by Lawrence07 (see Sec. 5 related work), all assume the same general form as in (4). The only difference is that they further assume \psi is linear, while ours do not have such restrictions. Do you imply all these pioneer works are wrong?
>
> C2: The paper itself acknowledges this by giving series of examples which all are functions of “f”
>
> R2: This is a misunderstanding. The definition of \psi only gives how we model the physical system. It has nothing to do with the definition of the extra source term. In general, all the equations, should have the from \psi[f]-g = 0. We just move g to the right.  In the ideal case, g can be constant 0 --- this is how most textbooks introduce various equations. But in practice, this is rarely true, because no system is completely closed.
>
> C3: the eq 6 states that a non-linear differential of a GP is some other Gaussian process. This is incorrect: non-linear transforms of a Gaussian process does not retain Gaussianity. Since this is a central definition of the paper, the whole method is likely incorrect.
>
> R3: First, we NEVER claim  a non-linear differential of a GP is some other GP’’, neither does Eq6 reflect it.  Eq6 only states given the posterior form of f, how we obtain g --- based on which we construct the conditional prior of \textbf{g} in Eq7, which obviously is NOT a GP. Second, in our modeling, we NEVER place a GP prior over g --- that will cause the double prior problem, and result in an INVALID probabilistic model. We use the symmetric property of Gaussian, to instead sample a set of virtual observations 0 given \textbf{g}, see Eq8 and the surrounding text. This is equivalent to a GP prior only in COMPUTATION, rather than from modeling! This is also our contribution --- we want to regularize the learning of g (and then f) with a simpler prior, i.e., GP, but still obey the Bayes’ rule to ensure proper, well-calibrated uncertainty quantification. Our model fulfills this effect in computation but in a correct probabilistic framework.
>
> C4: the paper introduces a bizarre concept of constant zero vectors as random variables, ie. p(0|g). This is clearly wrong, and the probabilistic model is then wrong as well” “Double priors are perfectly fine in Bayesian modelling, and eq9 could be defined without the 0-stuff
>
> R4: First, double priors will lead to an improper Bayesian model with an incorrect joint probability distribution. It might be fine if we just want point estimations and predictions. However, we aim to also improve uncertainty quantification in the posteriors. If the joint distribution is conceptually incorrect, how to justify the posterior computed based on the joint distribution?
>
> Second, we have clearly mentioned that we introduce a set of virtual observation to build a valid probabilistic model (see Sec1 &Sec3.1). This strategy is originally introduced in the Bayesian hybrid framework to principally combine a conditional and generative model (see Lasserre06 cited in our paper). The idea of virtual observations has also been used in Bayesian joint individual and group variable selection [1,2] and learning monotonic functions with GPs [3]. We do NOT see anything that is ``"Bizarre and clearly wrong".
>
> Third, regarding the product of experts (PoE), please see MacDonald’s15 at Sec3 end (cited in Sec5 of our paper), they make a clear comment about the advantage of an alternative model GPODE over the PoE based model AGM: ``Conceptually, the GPODE is a proper probabilistic generative model, which can be consistently represented by a directed acyclic graph (DAG)”.  This is consistent with the advantage claimed in our paper.
>
> [1] Feng, Y, and Qi, Y. "EigenNet: A Bayesian hybrid of generative and conditional models for sparse learning." NIPS, 2011.
>
> [2] Zhe, S, and et. al. Joint network and node selection for pathway-based genomic data analysis. Bioinformatics (2013), 29(16), 1987-1996.
>
> [3] Riihimäki, J, and Vehtari, A. "Gaussian processes with monotonicity information." AISTATS, 2010.

---

> > ### Comment · AnonReviewer3 · 2020-11-14
> > **ambiguities**
> >
> > Thanks for the clarifications. The eq 4 now makes sense. However, it should be revised to clarify that its not a function definition, but instead a \phi-g=0 equality. Also perhaps also explain what is g(x) in the burger example.
> >
> > I am still very confused on the probabilistic model.
> >
> > o So “f" is a GP, its differential is “h”, and it is equal to “g”. Then “g” is a function of a GP (linear or nonlinear).
> >
> > However, “g” is also not a GP (as stated above), but its still a Gaussian. I’m confused. What is g?
> >
> > o What does “prior only in computation, not modelling” mean?
> >
> > o What does “sample set of virtual 0’s” mean? Are you really repeatedly sampling vector of zeros? Sampling means that we draw values wrt their density. If we always sample a “0”, then we ignore densities.
> >
> > In [3] they use virtual observations, which are obviously ok. In [2] they use virtual zeros, and I am equally baffled what does it mean for them. [1] and Lasserre06 both try to combine generative and discriminative modelling, and both arrive at rather conventional hierarchical probabilistic models with no virtual zeros. I fail to see how any of these references justify or explain the use of zeros here. Virtual observations and virtual zeros are different things.
> >
> > The eq 8 has p(0|g,Z). This implies that “0” is a random variable with a point domain “0 \in {0}”. It also has a Gaussian prior, whose support is infinite. This is wrong. Can you clarify what the “0”-sampling, or what is p(0|..) or what is the “0” here?
> >
> > Even if the 0-sampling somehow would be theoretically ok (this would need solid justifications), I still find this to be a strange way of introducing non-standard notation and unnecessary complexity. If the model doesn’t work without these kind of hacks, something is wrong. No other paper in this domain have needed the 0-sampling, why its necessary here?
> >
> > o eqs 6 and 7 both relate the “g" (a random variable) to h(.,eps) (a non-random variable). What is the meaning of this?
> >
> > o what is the "double prior” problem here? To me it looks like one would get a product/decomposed prior (eg. p(g) = p_1(g) * p_2(g)), which is fine in probabilistic modelling.
> >
> > I would recommend the authors to revise the sec 3 completely, and make it clear and rigorous what is the model first (ie. what is g, what is h, what is f), and afterwards define how to sample from it. Mixing sampling and model definition is confusing. If these are mixed, it needs clearer justification.
> >
> > The author’s should also present the plate diagram for additional clarity.

---

> > > ### Author Response · Authors · 2020-11-16
> > > **clarification; we appreciate a careful reading of our paper to avoid ambiguity**
> > >
> > > C1: The eq 4 now makes sense. However, it should be revised to clarify that its not a function definition, but instead a \phi-g=0 equality.
> > >
> > > R1: We can certainty use \psi[f]-g = 0. However, it is pretty common/standard to use \phi[f] = g in the literature, not only in computational physics (see Raissi17), but also in prior GP works, see Graepel03 and LFM in Alvarez09 (all cited in Sec. 5 Related Work).
> > >
> > > C2: what is g(x) in the burger example.
> > >
> > > R2: In our application, Burger’s equation is used to describe the traffic flow in a highway; g(x) represents the external influences on the flow, e.g., new vehicles joining in or old vehicles leaving through the ramp. We will emphasize this point in the experiment.
> > >
> > > C3: However, “g” is not a GP, but it’s still a Gaussian. I’m confused. What is g? What does “prior only in computation, not modelling” mean?
> > >
> > > R3: “g” is a NOT Gaussian. The Gaussian is p(0|g, Z) in Eq. 8, which is used to sample virtual observations 0 given g (NOT to sample g itself). Note the joint probability (9) only includes p(0|g,Z) rather than p(g|Z) (in Eq.5).
> > >
> > > In Eq. (5) and the surrounding paragraph, we discussed IF we consider a GP prior for g, how this prior should be constructed. This is to motivate the generative component proposed later. In the following paragraph under Eq.7, we consider tying (5)&(7); however, “directly multiplying (5) and (7) is problematic” due to the double prior problem --- g has already been sampled according to f (see eq. 6&7), so we cannot assign a second prior over g (i.e., Eq. 5). To obtain a valid probabilistic model, we only use the first prior (i.e., Eq. 7) to sample g; then given g, we sample the virtual observations 0, i.e., p(0|g, Z). In our final joint probability in Eq. 9, there are NO terms that reflect g is a Gaussian.
> > >
> > > When we calculate the log probability for training, the log of p(0|g, Z) (generative component), i.e., logN(0|g, Sigma), is equal to computing the log of a GP prior over g with zero mean function, i.e., logN(g|0, Sigma) in Eq5. This is obvious, because Gaussian has the symmetric property --- switching the variable and mean does NOT change the density (see Eq. 8). Therefore, although in the modeling, we did not place a GP prior over g, in the computation (i.e., training), we equivalently calculate a GP prior of g.
> > >
> > > C4: What does “sample set of virtual 0’s” mean? …“Virtual observations and virtual zeros are different things.”… “Sampling means that we draw values wrt their density. If we always sample a “0”, then we ignore densities.”
> > >
> > > R4: First, we NEVER claim ``"virtual zeros” in our paper. What we claimed is “virtual observations 0”, see 1st line under Eq 8. In our experimental setting, we also mentioned the choice of the number of “virtual observations” (see Sec. 6.1 Competing Methods, and 6.2 Motion capture). We believe these clearly mean that we introduced a set of virtual observations, whose values are chosen to be zero. Note we have also discussed how to choose nonzero values (see 5th line  under Eq.8).
> > >
> > > Second, you might mix up random variables (RV) and their observations (instances). In essence, both our and the prior works [1-3], introduce a set of virtual, observable RVs. Let us denote them by s.  In our model, the conditional density is p(s|g, Z) = N(s|g, Z) (see Eq. 8). Note s are in continuous domain R^M.  In [3], the conditional density is a Probit likelihood given the latent GP derivatives. In our model, we assume the observed values of s are 0 while in [3] +1 or -1, in [1] the eigen-vectors of data covariance matrix, and in [2] also 0.
> > >
> > > Third, from the modeling structure, there are NO essential differences from choosing different values of the virtual observations. Such choices are totally determined by the problem/application. The key is that, using the hybrid framework, the model can avoid the double prior problem, and turn a second prior of the interested RV into a generative component to sample those virtual observations (conditioned on the RV). In this way, we can integrate an extra prior constraint/regularization in the training while the model is still in a principled Bayesian framework. That’s the motivation of the prior hybrid Bayesian models [1-3].
> > >
> > > C5: “what is the "double prior” problem here? To me it looks like one would get a product/decomposed prior (eg. p(g) = p_1(g) * p_2(g)), which is fine in probabilistic modelling”
> > >
> > > R5: Yes, you can alternatively claim to construct a new prior p(g) = p_1(g)*p_2(g). However, you must prove this is a valid prior distribution, i.e., \int p(g) = 1, p(g)>=0, etc. Simply multiplying Eq. 5&7 (Delta prior) will NOT give such a justification. We choose to turn p_2(g) into a generative component in the Bayesian hybrid framework.
> > >
> > > C6: The author’s should also present the plate diagram for additional clarity.
> > >
> > > R6: We DID provide a graphical model representation in Fig. 1 of the supplementary material. We mentioned it in Sec. 3.1, the end of the 2nd paragraph.

---

> > > > ### Comment · AnonReviewer3 · 2020-11-16
> > > > **ambiguities**
> > > >
> > > >
> > > > Thanks for the clarifications, the author intention is becoming much clearer. My current take is:
> > > >
> > > > * eq 5 does not hold, but the text should be clearer that this equation is not meant to hold. It’s quite unusual to give precise definitions of hypothetical terms.
> > > > * I see the motivation of the 0/g “trick”, since one wants to have an acyclic joint distribution. However, I still have concerns on the treatment of “0” being statistically correct. A new “virtual” random variable can certainly be introduced, but it has to be properly defined. My concerns:
> > > >     * The “0” is a random variable, which is sampled from a Dirac but evaluated with Gaussian density. I don’t follow what this means statistically, or whether the RV is Dirac or Gaussian. If one has a coin, but always gets “heads”, then the coin either (i) is not two-sided at all, or (ii) its density of {heads,tails} is {1,0}.
> > > >     * I not convinced the last equality of eq 8 is correct. A density “p” is always over some random variable, but I’m not sure what kind of RV the 0 is. The Gaussian density _function_ certainly is symmetric, but the Gaussian _density_ should not be (since the mean is not even a variable!). The “s” notation and intuition is very clear, thanks for the explanation. However in this case the “s” would again collapse to the constant zero, and hence not really work with the Gaussian of eq 8. It seems that in eq 8 the LHS 0 and RHS 0 are different zeros. In left its a constant, in right its a random variable. Overloading symbols should be avoided.
> > > >     * Finally, i’m unsure if the “trick” does anything. Eq 9 RHS is invariant to whether we have N(g|0) or N(0|g) density _function_, but it should not be invariant to whether we have p(g|0) or p(0|g) _density_. If the computing does not change at all, then surely there are also no modelling/theoretic changes either. I feel the authors are trying to apply an algebraic shortcut to acyclify their model without actually changing their underlying model, or fully accounting the effects of declaring the “0” as a _new_ random variable. To me the eq 9 RHS is still equivalent to the "double prior” case with p(g|0,S), but the necessary normalization constant of product priors p1*p2/Z is missing. On the other hand, the eq 9 LHS should not result in a Gaussian, but Dirac density for “0”.
> > > >         * Can you clarify whether the N(g|0)/N(0|g) choice changes something in practise or in computation?
> > > > * I’m having difficulty following eqs 6-7 or confirming that they are correct. Eg. in eq 6 we define g(.) to be a function of \eps, while g(.) does not contain \eps. I believe eq 6 tries to define a sample from p(g), but the notation is misleading in that case (the equality symbol should not be used for sampling, instead use \sim and eg. g^{j}). More generally, eqs 6-7 mix operations beteen RVs and samples (non-RV). Can you re-describe what eqs 6-7 mean? (see earlier comments as well)
> > > > * The joint probability eq 9 looks like a conventional hierarchical model. Can you explain why we need the sampling to be included here?

---

> > > > > ### Author Response · Authors · 2020-11-17
> > > > > **Clarification**
> > > > >
> > > > > C1: It’s quite unusual to give precise definitions of hypothetical terms Eq5.
> > > > >
> > > > > R1: Eq. 5 is important because it motivates our design of the generative component that has a different meaning in probabilistic sampling but is computationally equivalent. To present a reasonable logic flow, we need to first specify Eq. 5, and then explain why we cannot incorporate it outright to our probabilistic model, and why we should propose a generative component accordingly (see the following paragraph under Eq. 7).
> > > > >
> > > > >
> > > > > C2: The “0” is a random variable, which is sampled from a Dirac but evaluated with Gaussian density… not convinced the last equality of eq 8 is correct, not sure what kind of RV the 0 is…. in this case the “s” would again collapse to the constant zero …
> > > > >
> > > > > R2: The “0” here is NOT a random variable. It is the OBSERVATION of the virtual random variable (RV) “s” (see R4 in our previous response) and so it is data. The “0” is similar to the training inputs and outputs (X, y) (see Eq. 1& 9), which are observations of the input and output RVs, e.g., X = [0.3;0.4], y = [-1.3; 1.2]. These observations are fixed and used for training. In our work, we have an augmented dataset {(X, y), 0} to infer model parameters (similarly, all the prior Bayesian hybrid models [1-3] introduce augmented data).
> > > > >
> > > > > Second, the “0” is NOT sampled from Dirac. Given g, we have the conditional distribution of the virtual RV “s” as p(s|g, Z) = N(s|g, Z) (see Eq. 8 and R4 in our last response). Therefore, “0” --- as the observation (or value) of s --- is sampled from the Gaussian (only g is sampled from Dirac, see Eq. 7). N(0|g,z) --- RHS of Eq. 8 --- is the likelihood of this observation (more specifically, the likelihood of RV “s” taking “0”). If still uncomfortable, look at p(y|X) in Eq. 1, which is also a likelihood --- the likelihood of the output taking y give the input taking X.
> > > > >
> > > > > Third, “s” does NOT collapse to constant 0”. “s” is a continuous RV. Its range is R^M, NOT a singleton {0}. The “0” is the data (or observation or instance) for “s”. We combine the original data (X,y) and the virtual data “0” for model estimation. We can also construct nonzero virtual data/observations for “s”. See the discussion in our paper (5th line under Eq. 8).
> > > > >
> > > > > C3: Can you clarify whether the N(g|0)/N(0|g) choice changes something in practise or in computation
> > > > >
> > > > > R3: There is no change in computation. The change is the model definition.  N(g|0) defines another prior over g (in addition to Eq.7) and will cause a double prior problem, while N(0|g) defines a generative component that can be integrated into the original GP in a valid and principled way --- this is our modeling goal.
> > > > >
> > > > > C4: If the computing does not change at all, then surely there are also no modelling/theoretic changes either …
> > > > >
> > > > > R4: If this is true, there is NO meaning to propose the Bayesian hybrid framework by Lasserre06 and also the prior works [1,2]. Modeling and computation are obviously different things; you CANNOT use one thing to justify the other. Modeling specifies how you formulate the problem and defines the relationship of the variables. In Bayesian framework, modeling is defined by a sampling procedure and a joint probability distribution. Computation is the procedure to estimate/calculate the parameters and often involves optimization. They are verified in different ways. For example, in Bayesian modeling, we should ensure the sampling procedure is valid (DAG structure); in computing, we should look at if the (local) optimum is arrived or the Markov chain is mixed. One procedure cannot be used to verify the other. For example, two models have the same parameter estimations cannot prove that the two models have the same meaning.
> > > > >
> > > > > C5: I’m having difficulty following eqs 6-7 or confirming that they are correct … Can you re-describe what eqs 6-7 mean?
> > > > >
> > > > > R5: Eq. 6 describes how to draw a sample of g() (in the function space) conditioned on X and y in our model. First, according to the GP predictive distribution, we draw a sample of f given {X,y}: f() = \mu() + \epsilon\sqrt{v()} (see eq. 2 and paragraph containing Eq.6). Then according to the differential equation (4), we apply \psi[] to obtain the sample of g(), as given in Eq. 6. Note that function sample of g includes \epsilon.
> > > > >
> > > > > Eq. 7 (the Delta prior) actually evaluates the sampled g() (in Eq.6) at Z, to obtain the samples of [g(z_1), …, g(Z_M)], i.e., the finite projection of the function sample.
> > > > >
> > > > > The whole sampling procedure is as follows. Given X, we sample y (standard GP). Given X and y, we sample (epsilon and then) the values of g() at Z (eq. 7), i.e., \textbf{g}. Then given \textbf{g}, we sample the virtual observations “0” (eq. 8). This exhibits a DAE structure and leads a valid joint probability Eq. 9.
> > > > >
> > > > > C6: Can you explain why we need the sampling to be included in eq. 9?
> > > > >
> > > > > R6: The sampling procedure defines our model and results in the joint probability in Eq. 9. They have no conflicts. They are equivalent. See R5.

---

> > > > > > ### Comment · AnonReviewer3 · 2020-11-18
> > > > > > **more**
> > > > > >
> > > > > > Thanks for the clarifications, the dataset/generative/likelihood interpretation of 0 make a lot more sense. Clarifying this in the paper would be good (eg. use gray nodes for data variables in plate diagram).
> > > > > >
> > > > > > There’s some confusion on nomenclature here. In Bayesian modelling we have a joint distribution over data variables, model parameters and hyperparameters; for which a factorisation is chosen with the chain rule (-> DAG). The posterior is model parameters conditioned on rest, and (Bayesian) _inference_ is the approximation of the parameter posterior (eg. MCMC, MF-VI or MAP). _Model selection_ is the procedure to select hyperparameters (eg. MLL optim or CV).
> > > > > >
> > > > > > I would argue that the g|0 or 0|g choice _must_ give a different implementation, otherwise the double prior problem has no practical relevance at all. I believe the 0|g case is roughly correct, but the g|0 case would need an extra normalizing constant due to having a product prior (the product is not guaranteed to integrate to 1). The 0|g model on the other hand seems more or less ok.
> > > > > >
> > > > > > I believe what’s actually happening in eqs 6-7 is that g is a change-of-variables transform of the GP-f by the \phi. This properly defines g as a function of another RV, and also properly defines its distribution (via jacobian). If \phi is linear, then g is a GP.
> > > > > >
> > > > > > Using sampling to define a distribution is awkward. Using sampling instead to approximate a distribution is ok.
> > > > > >
> > > > > > Having a prior for g feels unnecessary. If f is smooth and \phi is smooth, then g=phi[f] will also be smooth. Having another smoothness prior for g is then unnecessary. The only cases where separate g-prior (or 0|g) makes sense are if we have separate actual observations directly from g(x), or if we can’t compute \phi. I’m not sure if either applies here.

---

> > > > > > > ### Author Response · Authors · 2020-11-19
> > > > > > > **Clarification**
> > > > > > >
> > > > > > > C1: use gray nodes for data variables in plate diagram
> > > > > > >
> > > > > > > R1: Great suggestion. We will add grey colors in our graphical representation.
> > > > > > >
> > > > > > > C2: use 0|g must give a different implementation because but the g|0 case would need an extra normalizing constant due to having a product prior (the product is not guaranteed to integrate to 1).
> > > > > > >
> > > > > > > R2: Yes, we totally agree. In R3 of the last response, we thought you were asking if simply multiplying Eq. 7 with Eq. 5 (i.e., N(g|0)) will lead to a difference in computation. In that case, the computation will be the same as in Eq.9. However, if we want to convert the product of the two priors into a new, valid prior, so that it can be present in a correct joint probability distribution (like Eq. 9), we do need an extra normalizer. This normalizer is only constant to g and will vary with the kernel parameters in Eq. 5 and equation parameters in Eq. 4 (if there are). Therefore, the computation will be very different.
> > > > > > >
> > > > > > > C3: what’s actually happening in eqs 6-7 is that g is a change-of-variables transform of the GP-f by the \phi. This properly defines g as a function of another RV, and also properly defines its distribution (via jacobian). If \phi is linear, then g is a GP.
> > > > > > >
> > > > > > > R3: We agree. We also discussed how to obtain the marginal distribution of g via Jacobian (see the text right under Eq. 7 in our main paper and Sec. 2 of the supplementary material). While in theory feasible, the computation is tricky in practice. So, we use the full joint distribution in Eq. 9.
> > > > > > >
> > > > > > > C4: Using sampling to define a distribution is awkward. Using sampling instead to approximate a distribution is ok.
> > > > > > >
> > > > > > > R4: We use a sampling procedure that describes how each random variable is in turn sampled or generated (until the observations/data are generate), to define our probabilistic model. Then we proceed to the joint distribution of the model. We believe this is pretty natural and intuitive. Actually, many classical papers, like Latent Dirichlet Allocations [1] and its extension, e.g., [2] (see Sec. 3 in both [1] and [2]), use the same way to introduce their models. Note that the sampling procedure to describe our model is DIFFERENT from the Markov-Chain Monte-Carlo sampling algorithm used to approximate the posterior.
> > > > > > >
> > > > > > > [1] Blei, David M., Andrew Y. Ng, and Michael I. Jordan. "Latent dirichlet allocation." JMLR, 2003: 993-1022.
> > > > > > >
> > > > > > > [2] Griffiths, Thomas L., et al. "Hierarchical topic models and the nested Chinese restaurant process." Advances in neural information processing systems, 2004.
> > > > > > >
> > > > > > > C5: Having a prior for g feels unnecessary. If f is smooth and \phi is smooth, then g=phi[f] will also be smooth ...
> > > > > > >
> > > > > > > R5: Our goal is to use Eq. 4, i.e., \psi[ f ] = g, to guide or facilitate the learning of f. We neither know the form of f nor the form of g.  All we know is their relationship \psi[]. Therefore, we put a f-prior and g-prior (or 0|g), and then connect them via \psi (our whole model is talking about how to build this connection in a valid probabilistic framework). In this way, we can use the g-prior (or 0|g) to regularize the learning of f. While g-prior might not be that informative, the regularization itself has encoded the valuable physics knowledge exhibited in \psi (see the paragraph right above Sec. 4.2) and will benefit f’s learning, as demonstrated in our experimental results.
> > > > > > >
> > > > > > > Our strategy is in high level consistent with the latent force model by Alvarez09. They also assume Eq. 4 and unknown f and g. The difference is that they further restrict that \psi is linear and has an analytical Green’s function. They put a prior over g, and then derive the prior for f via kernel convolution. This is essentially another way to build up the connection of the two priors (but only feasible for linear operators). Note that while g-prior is not that informative, the physics knowledge has been encoded in the connection to f (i.e., the convolved kernel) and then facilitates the training. See Sec.5 Related Work for more discussions.

---

### Official Review · AnonReviewer1 · 2020-10-28
**The framework is sound, but novelty seems questionable**

**Rating:** 5
**Confidence:** 4

**Review:**

This work proposes a deep Gaussian Process (GP) framework for data modeling informed by dynamical systems.
The analogous to the one of gradient matching, where a GP regression problem is penalised by constraints taking the form of differential equations acting on the GP itself. This study merges this framework to the scalable deep-GP framework of Wilson et al, and proposes an approximated inference setting in which the problem can be optimised through stochastic variational inference.
The proposed method is tested in several experimental scenarios and compared to standard kernel-based methods, and to the Latent Force Model of Alvarez and colleagues. The results show promising performances in terms of prediction and stability of extrapolation.

The proposed methodology contributes to the domain of gradient-matching and extends the classical approaches to allow for scalability in deep models. In this sense, the comparison with respect to the state of the art appears weak. The comparisons proposed in the study are with respect to either shallow models, or deep-models not allowing the integration of physics-informed contraints. Recent works (e.g. [1,2,3,4]) could provide a fairer benchmark for the proposed approach. In this sense, the feeling is that proposed methodology largely overlaps with these more recent studies, for example for what concerns the use of deep models and variational inference schemes. In particular, the idea of soft-regularisation of deep-GP models has been already explored in [2], and the authors may want to compare the method with respect to this approach.

It is not clear whether the parameters of the dynamical systems in the term h(Z,\epsilon) of (12) are inferred. If this is the case, the ability of the framework in estimating the systems’ should be assessed and compared to other moment matching approaches. This aspect is overlooked in the paper, although it is quite relevant for interpretability purposes.

---

> ### Author Response · Authors · 2020-11-14
> **thanks for your comments and a few clarifications**
>
> C: comment; R: response
>
> C1: “This work proposes a deep Gaussian Process (GP) framework for data modeling informed by dynamical systems. … In this sense, the comparison with respect to the state of the art appears weak… Recent works (e.g. [1,2,3,4]) could provide a fairer benchmark for the proposed approach”
>
> R1: Actually, our model is not a deep GP as originally proposed in Damianou13. We did not stack many GPs in several layers. We only have one GP, but the kernel is constructed from a (deep) neural network. We aim to use physics to regularize the estimation of the deep kernels so as to enhance the GP learning. That’s why in the experiments, we only compare with state-of-the-art single GP approaches. Incorporating physics into deep GPs is definitely promising and we will consider it as our future research plan.
>
> Second, while our evaluation includes ODEs, our model is not restricted by being informed by dynamic systems. Eq. (4) is general and NOT necessarily an ODE; it can be any PDE with or without time variables.  See Sec. 6.2 metal pollution prediction where we actually did not use any time information.
>
> Third, from the review, we cannot find the references numbered as [1,2,3, 4]. We will appreciate if the reviewer could list the corresponding papers. We will definitely double check.
>
>
> C2: “It is not clear whether the parameters of the dynamical systems in the term h(Z,\epsilon) of (12) are inferred. If this is the case, the ability of the framework in estimating the systems’ should be assessed and compared to other moment matching approaches”
>
> R2: Great question. First, as mentioned in Sec. 3.1, the 3rd line under Eq. (4), we allow \psi[] to include unknown parameters. Therefore, h couples the deep kernels, the differential operators, and unknown parameters in the differential equation. The parameters will be jointly estimated in the training. We will emphasize this point in our paper.
>
> Second, it is a great idea to assess these learned parameters. We will supplement the assessment. However, the goal of our paper is NOT system identification (via GPs). Rather, we aim to use incomplete differential equations to improve deep-kernel GP when learning with scarce data, in both prediction accuracy (esp. extrapolation) and uncertainty quantification. On two synthetic datasets and four real-world applications, our method has clearly shown improvement in both aspects. Therefore, we believe the advantage of our method has been well demonstrated. While many excellent works (e.g., gradient matching, see the references in Sec. 5) were proposed to (accurately) recover the equation parameters, they demand we have sufficient samples in the domain and restrict to special type of equations (typically ODEs). Most of them also require the equation form is completely specified. The problem setting is hence very different from ours, where the data is scarce, the equation is arbitrary, and include unknown source terms (functions).

---

> > ### Comment · AnonReviewer1 · 2020-11-16
> > **literature and benchmarks**
> >
> > I apologise for the missing references, this was due to a bad copy-paste during the writing of my reviews.
> > Please below you can find the above mentioned papers. It is worth mentioning that much of this recent literature deals with the problem of scalability in GP, deep, and probabilistic models accounting for constraints, not limited to dynamical systems. Moreover, the focus of these work is not necessarily on model identification, but also on uncertainty quantification and predictive accuracy.
> >
> > [1] Gorbach, N. S., Bauer, S., and Buhmann, J. M. Scalable Variational Inference for Dynamical Systems. In Guyon, I., Luxburg, U. V., Bengio, S., Wallach, H., Fergus, R., Vishwanathan, S., and Garnett, R. (eds.), Advances in Neural Information Processing Systems 30, pp. 4809– 4818. Curran Associates, Inc., 2017.
> >
> > [2] Lorenzi, M. and Filippone, M.. Constraining the Dynamics of Deep Probabilistic Models. In: International Conference on Machine Learning. 2018. p. 3227-3236.
> >
> > [3] Wenk, P., Gotovos, A., Bauer, S., Gorbach, N. S., Krause, A., & Buhmann, J. M. (2019, April). Fast gaussian process based gradient matching for parameter identification in systems of nonlinear odes. In The 22nd International Conference on Artificial Intelligence and Statistics (pp. 1351-1360). PMLR.
> >
> > [4] Pan, S., & Duraisamy, K. (2020). Physics-informed probabilistic learning of linear embeddings of nonlinear dynamics with guaranteed stability. SIAM Journal on Applied Dynamical Systems, 19(1), 480-509.

---

> > > ### Author Response · Authors · 2020-11-18
> > > **These methods cannot apply to our problem for comparison**
> > >
> > > Thanks for these great references. We will definitely cite and discuss them. However, all these methods CANNOT apply to our problem --- incomplete differential equations (with a latent source function) and scarce data --- and work as proper baselines.
> > >
> > > [1] inherits the product-of-expert (PoE) heuristic of the AGM model (see Calderhead09 and Macdonale15 cited in Sec. 5 Related Work of our paper) and marginalizes out the derivates of the target function in ODEs. As a consequence, [1] and AGM only apply to ODEs with explicit forms. You cannot introduce an unknown source function in the ODE (like the one in Sec. 6.1 of our paper) --- otherwise, the marginalization is NOT doable (see Eq.9 in [1]). Furthermore, [1] further restricts the ODE response to be a summation of the product of states (i.e., mass-action kinetics, see eq. 10 in [1]) to ensure a closed-form update in their mean-field variational inference. Therefore, the method [1] CANNOT deal with incomplete, general differential equation in Eq.4 of our paper (even \psi corresponds to an ODE).
> > >
> > > [3] discusses the issue of the PoE heuristics in AGM and [1]: the correct normalization will disconnect the observed data and ODE parameters. [3] provides another graphical model that can justify the previous approaches that uses an improper normalization to retain the connection. [3] also proposes a hybrid of point estimation (for GP hyperparameters) and MCMC (for hidden states and ODE parameters) for inference. However, all these discussions and extensions are based on the SAME setting as in AGM. That means, [3] is strictly grounded on ODEs with explicit forms, and cannot introduce latent source functions.
> > >
> > >
> > > While [2] can incorporate general equality or inequality constraints (based on ODEs/PDEs), it strictly requires every function in the constraint has an explicit form (e.g., H_{hi} listed in Sec 3.1 of [2]); otherwise, the constraint likelihood (e.g., Eq. 4&5 in [2]) CANNOT be constructed. The reason is that they need to evaluate both the LHS and RHS of the equality/inequality (on a set of inputs) to obtain the constraint likelihood in inference.  Therefore, their method does NOT support unknown source functions (i.e., g() in Eq 4 of our paper) as well, and hence is NOT applicable to our problem.
> > >
> > > While very interesting, [4] is the most IRRELEVANT to our work, in both the setting and the goal. First, [4] only focuses on (nonlinear) dynamical systems (i.e., ODEs) (our work is for general differential equation). Second, [4] assumes either the ODE is completely known (i.e., differential form in Sec. 2.2 in [4]) or ODE is completely unknown but there are enough, discrete trajectory data (i.e., recurrent form). By contrast, our model assumes the DE is partially known, i.e., with explicit \psi but an unknown source function g() (see Eq. 4) and the data is scarce. Therefore, even only considering ODEs, our setting does not apply to [4], and either is infeasible to the method in [4] (differential form) or leads to very bad results due to insufficient data (recurrent form). Third, [4] aims to learn a set of bases functions (with neural networks) that span the Koopman invariant space. These bases functions decompose the solution function to facilitate the analysis of the nonlinear dynamic system. This is far different from our goal that uses incomplete differential equation to enhance the learning of the target function (esp. for extrapolation) from scarce data. We do NOT see any common metric that we can use to compare the two works.

---

> > > > ### Comment · AnonReviewer1 · 2020-11-20
> > > > **Thanks for the clarification**
> > > >
> > > > I thank the authors for the clarification of their contribution in view of the recommended literature.
> > > > I acknowledge that the proposed approach may find broader applications than the ones possible with the methods I pointed out. As stressed by the authors, this is the case when the function g() is an unknown latent function.
> > > >
> > > > There are however comparisons considered in the paper (e.g. experimental scenarios in sections 6.1 and 6.2) where the comparison would be definitely possible (for example with respect to [2]) , and probably quite insightful as well, as it would allow the benchmarking with respect to more flexible modeling approaches (deep Bayesian architectures).

---

> > > > > ### Author Response · Authors · 2020-11-23
> > > > > **Thanks much for your comments and a bit more clarification**
> > > > >
> > > > > First, we want to emphasize that our goal (and also the contribution) is to integrate INCOMPLETE differential equations, \psi[f] = g where g is unknown, into deep kernel learning. Part of the reason is that there have been many excellent works proposed to exploit complete equations (i.e., g is known), including not only the great references [1-4], but also the ones cited in our paper (see Sec. 5 Related Work) as well as the recent physics informed deep neural networks (PINN), which directly motivates our work (see Sec. 1 Introduction 3rd paragraph). Actually, our setting is the same as the latent force models (LFM) by Alvarez09, except that LFM further restricts \psi to be linear (while we do not). That’s why in our experimental design, we evaluate the incomplete equation case (i.e., g is unknown) to match the goal of our work.
> > > > >
> > > > > Second, we do agree that in Section 6.1 and 6.2, if we disclose the form of g to the competing methods, we can apply [2] for learning as well, although this will be UNFAIR to LFM and our method. We will conduct such evaluations and see the results.
> > > > >
> > > > > Finally, thanks for suggesting the deep Bayesian architectures, which is very promising. We will definitely consider extending our model along this direction.

---

### Official Review · AnonReviewer4 · 2020-10-30
**Positive review for Physics Informed Deep Kernel Learning**

**Rating:** 8
**Confidence:** 4

**Review:**

The paper is clearly written, technically sound and innovative.

### Impact:
The paper is bringing an important contribution in the domain of Physics-Informed Machine Learning. It proposes an interesting and technically sounds (and not obvious) way to combine multiple successful components of recent literature in the field.

### Clarity and technical soundness:
The paper is clear, technically sound and manipulates tools in an advanced way.
It would be useful to provide more intuition and high-level interpretation on the most technical aspects of Deep Kernel inference and on the usage of ELBO methods. That would go at the benefit of understanding for the non-specialists of Kernel-based learning who are interested in Physics Informed ML.

### Results:
The results are extensive, are exploiting both simulation-type data and real-world data.
The benchmark used for comparison are relevant.

### Applicability:
It would be interesting to have a more high-level interpretation and analysis of the applicability of the method to 3D simulation data.
In 4.2, it was not clear to me what N and M stand for. Could you clarify it and provide a more high-level analysis with it too?

---

> ### Author Response · Authors · 2020-11-14
> **Thanks for your comments**
>
> C: comment; R: response
>
> C1: “In 4.2, it was not clear to me what N and M stand for”
>
> R1: N is the number of training examples and M is the number of latent inputs Z. We will polish our paper to be clearer.
>
> C2: “provide more intuition and high-level interpretation on the most technical aspects of Deep Kernel inference and on the usage of ELBO methods”
>
> R2: Great suggestion. We will definitely polish our paper and provide more intuition and interpretation.

---

### Decision · Program_Chairs · 2021-01-07
**Final Decision**

**Decision:**

Reject

**Comment:**

The paper presents a framework for incorporating physics knowledge (through, potentially incomplete, differential equations) into the deep kernel learning approach of Wilson et al. The reviewers found the paper addresses an important problem and presents good results.  However, one of the main issues raised by R1 is that, although the proposed method can be applied to broader settings such as that of incomplete differential equations, there are still regimes where the comparison is not only possible but perhaps insightful. An example baseline is the work of Lorenzi and Filippone, “Constraining the Dynamics of Deep Probabilistic Models” (ICML, 2018). Another critical issue, raised by R4, is the insufficient clarity in the presentation. Many of the concerns raised by this reviewer were clarified in the discussion and I thank the authors for their engagement. However, the AC believes some of the points raised by R4 in this regard were left unaddressed in the paper and the manuscript does indeed require at least one more iteration.

The format violation concerns raised during the reviewing process did not affect the decision on this paper, as the PCs confirmed that they did not meet the bar for desk rejection and recommended to assess the paper on its technical merits.